# Learning from labeled images and unlabeled videos for video segmentation

## Abstract

Performance on video object segmentation still lags behind that of image segmentation due to a paucity of labeled videos. Annotations are time-consuming and laborious to collect, and may not be feasibly obtained in certain situations. However there is a growing amount of freely available unlabeled video data which has spurred interest in unsupervised video representation learning. In this work we focus on the setting in which there is no/little access to labeled videos for video object segmentation. To this end we leverage large-scale image segmentation datasets and adversarial learning to train 2D/3D networks for video object segmentation. We first motivate the treatment of images and videos as two separate domains by analyzing the performance gap of an image segmentation network trained on images and applied to videos. Through studies using several image and video segmentation datasets, we show how an adversarial loss placed at various locations within the network can make feature representations invariant to these domains and improve the performance when the network has access to only labeled images and unlabeled videos. To prevent the loss of discriminative semantic class information we apply our adversarial loss within clusters of features and show this boosts our method's performance within Transformer-based models.

## 1 Introduction

Video object segmentation is attracting more attention due to its importance in many applications such as robotics and video editing. Much progress has been made on general video understanding tasks such as action recognition, but is lagging on dense labelling tasks such as video segmentation. This is mainly due to the laborious and time-consuming nature of pixel annotation collection, resulting in small, sparsely annotated video datasets. To deal with this problem, researchers often use large-scale image segmentation datasets to learn semantic representations through pretraining.

While image segmentation datasets are a useful source of labeled data for pretraining, the representations learned do not translate well to videos containing video artifacts such as motion blur, low lighting and low resolution. Figure 1 shows an example of the domain differences arising in common datasets, where the boundaries and small spokes of the moving bicycle are blurred. When image-pretrained models are applied directly to videos, a performance drop is observed in object detection (Kalogeiton et al., 2015; Tang et al., 2012) and in our own video segmentation experiments. Further, Kalogeiton et al. find that in addition to motion blur, the location of objects in the frame, diversity of appearance and aspects, and camera framing all have an effect on the performance of image-pretrained models on videos. Thus supervised training on images is insufficient for pixel-wise video understanding, producing a need for a convenient alternative representation learning method that uses unlabeled videos.

Unsupervised video representation learning has mainly focused on the classification task rather than on segmentation. Some methods learn useful features from unlabeled videos with data augmentations (Behrmann et al., 2021), contrastive losses (Han et al., 2021), and pretext tasks such as frame shuffling (Xu et al., 2019). The learned networks are successful on video classification, but do not translate well to segmentation due to a loss of detail that occurs in the spatial bottleneck. Applying the same self-supervised methods for segmentation leads to poor performance because the features learned do not contain enough local information for the decoder to reconstruct the output.

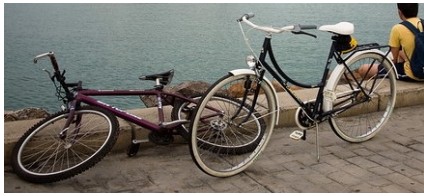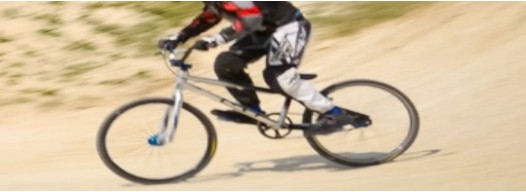

Figure 1: A single image from the COCO dataset (left) and a frame from Davis 2019 (right) shows domain differences such as motion blur arising from video artifacts.

In this paper, we propose an approach to video segmentation that takes advantage of both labeled image segmentation data and unlabeled videos. Taking inspiration from (Tang et al., 2012), we use unlabeled videos to minimize the domain difference between image representations and the spatial component of video representations. We train our video segmentation networks to be invariant to properties specific to video (motion blur, viewpoints, etc.) so that we can train on labeled images and apply them to videos without a performance drop.

To achieve this invariance we propose inserting a domain discriminator within a video segmentation network to discourage it from learning image or video-specific features. We train the network for segmentation using labeled images while using unlabeled videos to adversarially train the discriminator predicting the domain of the sample, shown in Figure 2.

We experiment with two different segmentation backbones: convolutional neural networks (CNNs) and Transformer-based networks. To take advantage of temporal information in unlabeled videos while retaining spatial information from labeled images, we also apply our method to VideoSwin (Liu et al., 2021b) with a spatiotemporal window size. In our CNN we place the discriminator at the end of the encoder prior to the decoding stage. In our Transformers we experiment with placing the discriminator at either the end of the encoder or after the patch embedding layer. We find that placing the discriminator after the patch embedding to target low-level features boosts the contribution of the adversarial loss. We conduct experiments using the video object segmentation (VOS) datasets Davis 2019 and FBMS and show that in our target setting with no access to labeled videos, our method improves segmentation performance over models supervised with images.

## 2 RELATED WORK

For video object segmentation a network must generate a pixel-wise classification for one or more moving target objects in a video. Labels must be consistent across frames but not necessarily across videos. In the unsupervised VOS track no annotations are provided during inference. Without the target object annotation from the first frame, unsupervised VOS methods rely on knowledge from image pretraining or optical flow. Li et al. (2018) leverage embeddings from an instance segmentation network trained on still images to generate an embedding for each object in a scene and then use semantic score and motion features from optical flow to select foreground object embeddings for a track. Similarly, RTNet (Ren et al., 2021) distinguishes foreground objects from moving distractors using a module that computes similarities between pairs of motion and appearance features from different objects. Recently, TransportNet (Zhang et al., 2021) aligns RGB and flow features in a two-stream network by optimizing Wasserstein distance with a Factorized Sinkhorn method. Yang et al. (2021c) do this alignment using co-attention between RGB and flow. Yang et al. (2021a) rely solely on optical flow as the input to their segmentation network. Optical flow features can be cumbersome to generate prior to segmentation so in our work we focus on learning strong semantic object representations from labeled images and unlabeled videos.

Much work has been done on self/un-supervised learning for classification, but video segmentation representations have not been as well explored. Inspired by MoCo (He et al., 2020), VideoMoCo (Pan et al., 2021) adopts a momentum encoder but drops frames from videos to improve temporal robustness. Vi2CLR (Diba et al., 2021) learns features from unlabeled videos within separate 3D and 2D CNN encoders by clustering latent frame and clip features then applying a contrastive loss between clusters. Our method differs from these because it targets learning detailed segmentation features. Self-training Zoph et al. (2020) is complementary to our method, as are self-supervised

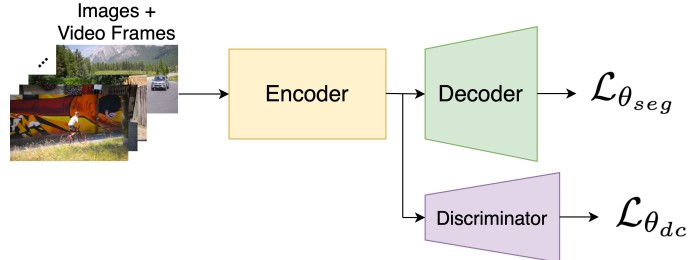

Figure 2: Our general adversarial method takes the output of the encoder and uses its feature representation to classify the training samples' domain. The discriminator's parameters are optimized by the adversarial loss.

objectives for video correspondence learning in video instance segmentation Fu et al. (2021). Our method differs from these in the way it uses video representations.

Many works have addressed domain adaptation for semantic segmentation in order to solve the synthetic-to-real problem or transfer representations. Shin et al. (2021) use distillation to tackle synthetic-to-real domain adaptation for video semantic segmentation. Guan et al. (2021) enforce similar temporal consistency between consecutive real frames and consecutive synthetic frames. Hong et al. (2017) use unlabeled videos for image segmentation by generating pseudo-labels. Tang et al. (2012) address image-to-video domain adaptation in object detection by retraining an image-trained detector on target video samples. Our work is most similar to that of Yang et al. (2021b), which uses domain adversarial learning on the class tokens of a vision transformer for image classification.

## 3 METHODS

### 3.1 DOMAIN ADVERSARIAL LOSS

Our main approach relies on treating images and videos as separate domains and applying a domain adversarial loss to learn robust object representations. Many image and video segmentation networks use encoders that learn low-level features in the shallow layers and abstract features in deeper layers. Based on observations by Kalogeiton et al. (2015), we hypothesize that the object representations learned by an encoder trained only on images differs from the representations learned on videos. This is due to video artifacts, especially motion blur, camera framing, appearance and aspect diversity. We further hypothesize that this makes it more difficult for a segmentation decoder to recover fine-grained details in videos. Our goal is to bring spatial features learned from images closer to those learned from videos using an adversarial loss propagated through the encoder. In a real-world scenario the most likely setting is access to a small amount of labeled images (and perhaps labeled videos) and a large amount of unlabeled videos. Our method makes it straightforward to use all available data.

To make the feature representations invariant to the image and video domains, we add a domain discriminator that distinguishes between the domains at different points in the encoder. During training we maximize the discriminator's loss, which encourages the encoder to learn feature representations that fool the discriminator by containing as little information as possible about the domain.

Formally, we are given samples from a source dataset $D^s = \{(x^s, y^s)\}_{s=1}^N$ where each $x^s$ is a single image and $y^s$ is its pixel-wise segmentation label. In the unsupervised domain adaptation setting we have access to samples from an unlabeled target dataset $D^t = \{x^t\}_{t=1}^M$ where each $x^t$ is a video frame. Given a sample $x_i \in \{D^s, D^t\}$, our discriminator estimates its domain $D_i$ through

$$\underset{D \in \{D^s, D^t\}}{\arg \max} \ \Pr(D_i = D \mid h(x_i)) \tag{1}$$

where $D_i$ is the domain of $x_i$ and $h(\cdot)$ is a function mapping $x_i$ to some latent feature space. In our case, $h(x_i)$ is the encoder's feature representation of $x_i$. Using a binary cross entropy loss, the

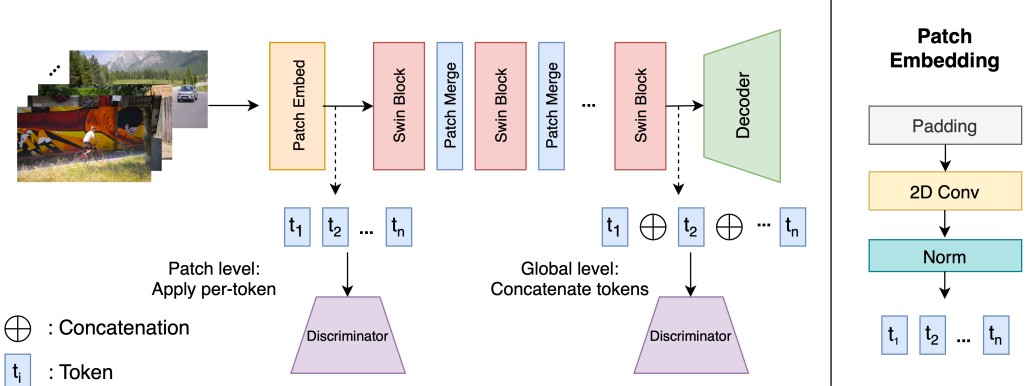

Figure 3: We place the adversarial loss after the patch embedding layer or at the end of the encoder. The patch embedding consists of convolutions applied per-patch. Here it is feasible to apply the discriminator per-token, allowing the discriminator to notice video artifacts at moving object boundaries which may not be present at deeper layers. At the global encoder level the tokens are concatenated then passed to the discriminator. Each *blue block* indicates a token $t_i$, with $n$ tokens in the sequence. $\oplus$ indicates token concatenation.

discriminator's training objective is to minimize

$$\mathcal{L}_{\theta_{dc}} = \sum_{(x_i, d_i) \in D} -d_i \log(p_i) + (1 - d_i) \log(1 - p_i) \tag{2}$$

where $\theta_{dc}$ are the parameters of the discriminator, $D = D^s \cup D^t$, $d_i$ is a binary indicator $\mathbb{1}[x_i \in D^s]$, and $p_i$ is the network's estimate of $\Pr(D_i = D^s | h(x_i))$.

During joint training with the rest of the segmentation network, a Gradient Reversal Layer (Ganin et al., 2016) between the encoder and discriminator. The layer outputs the identity on the forward pass and reverses the sign on the discriminator's gradients during the backward pass.

The encoder and decoder are trained for segmentation through a standard cross entropy loss:

$$\mathcal{L}_{\theta_{seg}} = \sum_{(x_i, y_i) \in D^s} \sum_j \sum_k y_{i,j} \log p_{i,j,k} \tag{3}$$

where $j$ iterates over each pixel in the image, $c$ ranges over the number of object classes and $p_{i,j,k}$ is the decoder's estimate of the probability that pixel $j$ in image $i$ belongs to class $k$. In case of spatiotemporal input, $j$ ranges over the temporal dimension as well. The final network is trained end-to-end by minimizing the segmentation loss and maximizing the adversarial loss

$$\mathcal{L}_{final} = \mathcal{L}_{\theta_{seg}} - \mu \mathcal{L}_{\theta_{dc}} \tag{4}$$

where $\mu$ is a constant scaling factor chosen through grid search to ensure the discriminator does not collapse. Our joint training procedure ensures that the network learns feature representations that are useful for segmentation and do not contain domain information. The general architecture of our method with loss components is shown in Figure 2.

## 3.2 TOKEN COMBINATION

We consider two types of segmentation backbones: CNNs and Transformers. In the CNN backbone, we place the domain discriminator at the end of the encoder, and flatten the spatial dimensions of the features for the discriminator's input. In a Transformer-based encoder the tokens may be passed individually to the discriminator or combined through concatenation. The discriminator can also be applied at any layer within the transformer whereas in the CNN the feature size in earlier layers is computationally expensive. The number of tokens at the end of the Transformer encoder is large so applying the discriminator per-token at this location would increase the loss's magnitude and dominate training. Instead we batch tokens within a sample and flatten their spatial dimensions. Figure 3 illustrates this token concatenation.

Some video artifacts such as motion blur are most apparent around moving object boundaries. When applied to tokens at the global level, the adversarial loss operates over the entire spatial resolution and may not target these features well. To address this we place our adversarial loss directly after the patch embedding layer to operate over low-level features containing detailed spatial information. We apply the loss per-token right after the patch embedding layer.

## 3.3 CLUSTERED ADVERSARIAL LOSS

We want our adversarial loss to make network features invariant to image-video domain differences while retaining discriminative class information. However our general adversarial loss formulation is agnostic to semantic class. This becomes an issue with sparse annotations: different image and video datasets may not be labeled with the same classes, but we want to take advantage of all labeled data regardless of annotation consistency.

To this end we propose grouping tokens from a batch containing image and video frames into class clusters then applying the adversarial loss within each cluster. At the patch embedding level, each token is likely to be dominated by information from a single class because tokens are formed from local patches. Thus an unsupervised clustering method, k-means, is likely to contain image and video features from a single class. Applying the loss within a cluster ensures the loss is not removing discriminative class information.

We modify the discriminator to take a pair of features and estimate the probability that they belong to the same domain. Formally our clustered adversarial loss becomes

$$\mathcal{L}_{cl} = \sum_c \sum_{(x_i, x_i') \in B_c} -b_i \log(q_i) + (1 - b_i) \log(1 - q_i) \tag{5}$$

where $c$ ranges over each cluster, $B_c$ is a set of pairs of image and video features within a cluster, $b_i$ is a binary indicator of whether $x_i$ and $x_i'$ belong to the same domain $D$, i.e. $b_i = \mathbb{1}[x_i \in D, x_i' \in D]$, and $q_i$ is the network's estimate of $\Pr(x_i \in D, x_i' \in D)$. While the k-means clustering step is the most time-consuming operation, it is only carried out during training and inference time remains the same.

## 4 EXPERIMENT SETTINGS

### 4.1 DATASETS

To understand how our method changes object representations learned from videos we conduct experiments on subsets of Davis 2019 (Perazzi et al., 2016) and on FBMS (Ochs et al., 2014). We use COCO Stuff (Lin et al., 2014) as a source of labeled images, which has many overlapping classes with Davis and FBMS.

Davis is an unsupervised video object segmentation dataset with annotations for 60 training and 30 validation videos. The original annotations label foreground target objects with a unique ID that is consistent within a video but not across videos. For our experiments we process Davis annotations so that they are semantically consistent across videos in order to better measure our network's understanding of semantics rather than identification of moving objects. We focused on 10 classes in common with COCO that appear in both training and validation splits. Having an object appear in the training set allows us to analyze how its object representation changes when using our method: any difference in performance results from changes in the object representations as opposed to the model's ability to generalize to unseen classes. A breakdown of the number of video frames in each of the shared classes in Davis can be found in the supplementary material. The final Davis training set consists of 3,401 frames and 1,558 frames for validation. The COCO training set used with Davis has 75,377 frames and the validation set has 3,176 frames.

As an additional source of unlabeled video data, we use YouTube-BoundingBoxes (YTBB) (Real et al., 2017). It features bounding box annotations for 23 objects in about 380,000 videos sourced from public YouTube URLs. When decoding the videos we use 30 fps and skip videos that have been taken down by the uploading user. Similar to COCO we use classes shared with Davis. Vi-

sual inspection of YTBB reveals that the videos often feature motion blur, low lighting, and low resolution.

FBMS (Ochs et al., 2014) contains 59 video sequences with 353 annotated training frames and 367 annotated test frames. The content ranges across 19 object classes most of which are in common with COCO. We process annotations to be semantically consistent across videos. When training COCO with FBMS we draw images containing classes in common with FBMS, resulting in 80,279 training images.

## 4.2 IMPLEMENTATION

To analyze the domain shift between images and videos, we train segmentation networks jointly on images and videos. For our CNN we use Deeplabv3 (Chen et al., 2017) with a ResNet-101 backbone. The domain discriminator consists of a 2D convolution to reduce the feature size, then a fully connected layer, dropout, ReLU and the classification layer.

For our 2D Transformer we use the Swin-T Transformer backbone with a UperNet decoder (Liu et al., 2021a) with patch size (4, 4) and window size (7,7). The discriminator is implemented as a sequence of alternating linear and leaky ReLU layers. When the discriminator is placed at the end of the encoder the tokens are concatenated and spatial dimensions flattened. At the patch embedding level each token's spatial dimensions are flattened and the token and batch dimensions are combined to apply the discriminator per-token.

Our spatiotemporal model is VideoSwin with a Swin-T backbone. It uses patch size (1,4,4) and window size (4,7,7). The discriminator is implemented as a PatchGAN discriminator (Isola et al., 2017) with four layers.

In all our experiments we use the PyTorch learning framework. Images and video frames are resized to 256 x 256 and normalized. In our experiments with VideoSwin we resize inputs to 512x512 and clip length 4 for videos, zero-padding single images to the same temporal resolution. The segmentation cross entropy loss uses loss weighting calculated from DAVIS or FBMS. The 2D CNN is initialized with COCO-pretrained weights from the PyTorch model zoo. The Swin Transformer is initialized using ADE20k weights provided by the model's authors. We initialize the VideoSwin Transformer with the same ADE20k weights and replicate them across the temporal dimension where appropriate.

The adversarial loss's coefficient $\mu$ is set to constant 1.0 that performed better than a slow increase. When training with the clustered adversarial loss the number of clusters is pre-set to the number of classes in the video dataset. On one GeForce GTX Titan X GPU, our 2D Transformer with the general adversarial loss takes 365 seconds to perform an epoch training on Davis with batch size 16 and 60.1 GFLOPS for inference. Under the same settings the clustered loss takes 1312 seconds and 60.0 GFLOPS. Our 2D CNNs are trained with batch size 8 and learning rate 0.001 and our 2D Transformer is trained with batch size 32 and learning rate 0.0032. VideoSwin is trained using batch size 16 and learning rate 0.0001. We will publicly release our code upon publication acceptance.

## 5 RESULTS

### 5.1 2D CNNS ON DAVIS

We focus on the setting in which we have access to labeled images and unlabeled videos due to the challenging nature of collecting video annotations. To test the adversarial loss' contribution to learning detailed semantic representations, we place a discriminator at the end of a 2D CNN encoder which acts on global spatial feature representations. We train on video frames from Davis, images from COCO and also experiment with training on random unlabeled YouTube videos from YTBB. Unlabeled frames are used in the adversarial loss but not the segmentation task loss. For our baseline we use the 2D CNN trained only on labeled images.

Table 1 shows our results on Davis measured using mean Jaccard index or Intersection over Union (IOU) over all semantic classes. We find that in our target setting with no access to labeled videos, the adversarial loss boosts the performance of the baseline CNN. The adversarial loss even boosts performance on Davis when using random unlabeled YouTube videos to train the discriminator. Our

Table 1: Results on Davis from training a 2D CNN on Davis, COCO and YTBB. Adv indicates whether or not an adversarial loss was included during training. Labeled and Unlabeled columns indicate the label setting for the training set. Results are measured on Davis by averaging IOU over 10 classes plus background. The adversarial loss boosts performance over the baseline trained on labeled images and no videos.

| Adv | Labeled | Unlabeled | mIOU | Δ |
|---|---|---|---|---|
| ✗ | COCO | – | 32.3 | – |
| ✓ | COCO | Davis | 36.9 | + 4.60 |
| ✓ | COCO | YTBB | 37.7 | + 5.40 |
| ✗ | Davis | – | 32.2 | – |
| ✗ | COCO+Davis | – | 46.2 | – |

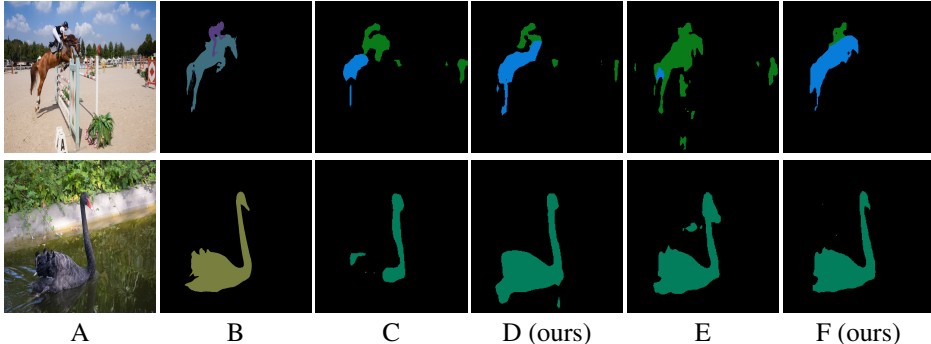

| A | B | C | D (ours) | E | F (ours) |

Figure 4: Qualitative results on Davis videos. *Column A:* The original frame. *Column B:* The ground truth. *Column C:* The 2D CNN results when trained on COCO (image data). *Column D:* The 2D CNN trained on labeled COCO and unlabeled Davis data with the adversarial loss. *Column E:* Swin results when trained on COCO images. *Column F:* Swin trained with the clustered adversarial loss on labeled COCO and unlabeled Davis frames. When the Swin transformer is trained with the clustered adversarial loss it recovers fine-grained details and has a better separation of semantic classes, whereas the models trained only on images show spurious and missing predictions.

method outperforms the supervised setting in which the model is trained only on labeled videos and no labeled images. Training with our adversarial loss on images and unlabeled videos performs better than training on just labeled videos, highlighting the usefulness of our method in settings where labeled videos are difficult to collect and object diversity is limited.

## 5.2 TRANSFORMERS

In the following experiments we test whether our adversarial loss boosts the performance of a Transformer-based model trained only on labeled images. We use Swin as our 2D Transformer model and VideoSwin with a spatiotemporal window size as our 3D Transformer.

**2D Transformer on Davis** We train a 2D Swin Transformer on videos from Davis and images from COCO and show results in Table 2. We find that placing the discriminator right after the patch embedding as described in section 3.2 yields better performance than applying it globally, showing that domain-specific features are better targeted at shallower layers. Training with the adversarial loss is able to improve performance in the unlabeled video setting as well as with access to labeled videos. We test the performance of our clustered adversarial loss and show that clustering tokens at the patch embedding level before applying the adversarial loss leads to a boost in performance.

**VideoSwin on Davis** We also explore how our method performs in a model that learns spatiotemporal features, namely, VideoSwin with image patch tokens and spatiotemporal window extending over the entire clip length. Table 3 shows the results of training this network using the general adversarial loss at the patch level. In the unlabeled video setting the adversarial loss improves on the

Table 2: Results on Davis from training a 2D Transformer on Davis and COCO. Adv indicates whether or not an adversarial loss was included during training. Cluster indicates whether the clustered or general adversarial loss was used. The Patch/Global column indicates the discriminator's location. Results are measured on Davis by averaging IOU over 10 classes plus background.

| Adv | Cluster | Patch/Global | Labeled | Unlabeled | mIOU | $\Delta$ |
|---|---|---|---|---|---|---|
| ✗ | – | – | COCO | – | 29.98 | – |
| ✓ | ✗ | Global | COCO | Davis | 25.27 | + 4.71 |
| ✓ | ✗ | Patch | COCO | Davis | 32.70 | + 2.72 |
| ✓ | ✓ | Patch | COCO | Davis | 38.53 | + 8.55 |
| ✗ | – | – | Davis | – | 25.97 | – |
| ✗ | – | – | COCO + Davis | – | 29.01 | + 3.04 |
| ✓ | ✗ | Global | COCO + Davis | – | 28.15 | + 2.18 |
| ✓ | ✗ | Patch | COCO + Davis | – | 32.55 | + 6.58 |
| ✓ | ✓ | Patch | COCO + Davis | – | 38.27 | + 12.30 |

Table 3: Results on DAVIS of VideoSwin with the general adversarial loss at the patch embedding level.

| Adv | Labeled | Unlabeled | mIOU | $\Delta$ |
|---|---|---|---|---|
| ✗ | COCO | – | 19.35 | – |
| ✓ | COCO | DAVIS | 30.48 | + 11.13 |
| ✗ | DAVIS | – | 25.75 | – |
| ✗ | COCO+DAVIS | – | 32.46 | + 6.71 |
| ✓ | COCO+DAVIS | ✗ | 33.65 | + 7.90 |

image-supervised model, showing that our method can be applied to spatiotemporal models that rely on image-pretrained weights.

**FBMS**   We test whether our method achieves a boost in performance on a different video dataset FBMS and show results in Table 4. In the unlabeled video setting, the general adversarial loss improves over the model supervised by images. Because the annotations for FBMS do not include all instances of each class in every video, there is a discrepancy between the annotations for FBMS and COCO which we believe prevents the model from learning strong semantic class representations, leading the model to perform worse when trained with both labeled videos and images than training with our adversarial loss.

## 6   ABLATION ON UNLABELED IMAGES

We conduct an ablation study to test whether the performance boost with our adversarial loss in the previous experiments is due to feature invariance in the image-video domains or invariance to dataset biases. To test this we train a 2D model on labeled images from COCO and unlabeled images from another large-scale image segmentation dataset Pascal VOC (Everingham et al., 2010) to train our adverarial loss. We use the entire Pascal dataset as a source of unlabeled data.

Table 4: Results on FBMS of training a Transformer with our general adversarial loss at the patch level.

| Adv | Labeled | Unlabeled | mIOU | $\Delta$ |
|---|---|---|---|---|
| ✗ | COCO | – | 33.98 | – |
| ✓ | COCO | FBMS | 34.47 | + 0.49 |
| ✗ | FBMS | – | 26.6 | – |
| ✗ | COCO+FBMS | – | 34.40 | – |

Table 5: Ablation study on whether the adversarial loss boosts improvement when training on labeled COCO images and unlabeled Pascal images. The adversarial loss is placed at the end of the encoder in a 2D CNN or Transformer. Adding unlabeled data from another image dataset does not significantly affect test IOU measured on COCO or Davis. This is in contrast to our image-video adversarial learning where training on unlabeled videos increased performance over the supervised image baseline 1 2.

| Model | Adv | Labeled | Unlabeled | Test | mIOU |
|---|---|---|---|---|---|
| CNN | ✗ | COCO | – | DAVIS | 32.3 |
| CNN | ✓ | COCO | Pascal | DAVIS | 33.7 |
| CNN | ✗ | COCO | – | COCO | 44.5 |
| CNN | ✓ | COCO | Pascal | COCO | 45.0 |
| Transformer | ✗ | COCO | – | DAVIS | 10.35 |
| Transformer | ✓ | COCO | Pascal | DAVIS | 10.21 |
| Transformer | ✗ | COCO | – | COCO | 11.31 |
| Transformer | ✓ | COCO | Pascal | COCO | 10.10 |

Our results in Table 5 show that using unlabeled images when training with the adversarial loss does not significantly improve results regardless of the backbone. Additionally the same observations hold when testing on COCO. We believe the adversarial loss' improvements are due to features becoming invariant to video artifacts, not object appearance diversity.

## 7 DISCUSSION

In experiments with the adversarial loss within a 2D Transformer, we found placing it at the patch embedding and applying the loss per-token increases performance over concatenating tokens at the end of the encoder and applying the loss over the concatenated feature. Each token at the end of the encoder contains information from the entire image. This makes it more difficult for the discriminator to target low level video features such as motion blur at moving object boundaries. Features at the beginning of the network retain fine-grained details that still have some separation of video-specific features and class information. While the patch-level loss can target low level feature differences in the domains, more abstract features present in either images or videos may be embedded in intermediate features. The model may benefit from placing the domain discriminator at intermediate blocks within the transformer. We leave this for future work.

Joint training of segmentation models on multiple datasets requires a significant overlap in semantic labels. Otherwise, missing information confuses the network and degrades performance. Our clustered adversarial loss retains discriminative class information by first clustering low level appearance features, typically aligned along class. Applying the discriminator within a cluster removes domain-specific but not class information. Additionally the loss does not rely on features specific to the image/video domain gap and could be applied to different adaptation tasks.

Finally, we note that while spatiotemporal learning has been critical for good performance in video classification, fully 3D end-to-end trainable networks for VOS have had limited success without relying on optical flow. Here we take a step towards developing a standalone 3D Transformer for VOS and show that our adversarial loss can improve on the supervised baseline.

## 8 CONCLUSION

In this paper we propose an unsupervised method to learn fine-grained representations from labeled images and unlabeled videos for video segmentation. It addresses a gap in the video representation learning literature where the focus is typically on video classification rather than segmentation. We leverage image segmentation datasets to learn strong semantic features then use an adversarial loss to address the performance drop that ensues when applying these networks to videos. Our findings indicate the general formulation of the loss improves CNN and 2D/3D Transformer's performance and our clustered adversarial loss improves 2D Transformer performance.

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
