# OpenReview forum: "Learning from Labeled Images and Unlabeled Videos for Video Segmentation"
_ICLR.cc/2023/Conference — Submitted to ICLR 2023_

### Official Review · Reviewer_KyqK · 2022-10-20

**Confidence:** 4
**Clarity, Quality, Novelty And Reproducibility:** The ideas are clear, novel and reprod…
**Correctness:** 3
**Technical Novelty And Significance:** 3
**Empirical Novelty And Significance:** 2
**Recommendation:** 5

**Strength And Weaknesses:**

Strength:
The idea of narrowing representation variance between image and video data is interesting.


Weakness:
1.The experimental results are not convincing.
1)Selection of the dataset is strange. The authors require large-scale video segmentation dataset with common classes in COCO, but they use the subset of the DAVIS 2019 dataset and the small-scale FBMS dataset, ignoring large ones like the Youtube-Objects Dataset which would be better qualified.

2)Performance for baseline methods on the DAVIS dataset is rather low, comparing with state-of-the-art results, e.g. 65.6% J mean on 2019-val by [1]. Effectiveness of the proposed loss function on such algorithms does not guarantee the effectiveness on top-performing ones.
[1]Luiten, J., Zulfikar, I.E., & Leibe, B. (2020). UnOVOST: Unsupervised Offline Video Object Segmentation and Tracking. 2020 IEEE Winter Conference on Applications of Computer Vision (WACV), 1989-1998.

2.The authors claim that ”Some video artifacts such as motion blur are most apparent around moving object boundaries. When applied to tokens at the global level, the adversarial loss operates over the entire spatial resolution and may not target these features well. To address this we place our adversarial loss directly after the patch embedding layer to operate over low-level features containing detailed spatial information. We apply the loss per-token right after the patch embedding layer.” in Section 3.2, but I cannot find the demonstration of this statement.


3.Experiment, performance not convincing.
There are many typos in this manuscript, some frequent errors are:
1)inconsistent verbal forms, e.g. sometimes ‘labelled’, sometimes ‘labeled’;
2)lack of commas, e.g. “To prevent the loss of discriminative semantic class information  (short of a comma) we apply our ...” in the abstract section, “To deal with this problem  (short of a comma) researchers often use ...” in the introduction section;
...

**Summary Of The Paper:**

This paper proposes an adversarial loss function between image and video domain to unsupervisely learn the fine-grained representations from labeled images and unlabeled videos for video segmentation. The authors show that their proposed method can improve the performance of both the CNN and transformer based methods on video segmentation.

**Summary Of The Review:**

The idea of this paper is interesting, but it is not well demonstrated.

---

> ### Author Response · Authors · 2022-11-18
> **Clarifying experiment details and performance**
>
> We thank the reviewer for their comments.
>
> > "Experiment, performance not convincing."
>
> First we would like to ask for clarification on what part of the performance is not convincing. We provide detailed settings for our experiments, and also plan to make our code available for reproducibility.
>
> > "The authors require large-scale video segmentation dataset with common classes in COCO, but they use the subset of the DAVIS 2019 dataset and the small-scale FBMS dataset, ignoring large ones like the Youtube-Objects Dataset which would be better qualified."
>
> Youtube-Objects has annotations for bounding box detection and does not come with segmentation annotations, making it less suitable. We chose Davis and FBMS in our experiments based on the availability of pixel-wise annotations for shared classes with COCO. For our method it is important that objects of the same class are consistently annotated between the image and video datasets; if this is not the case then it is possible for the discriminator to use the absence of a class to distinguish the domain, in which case the encoder learns not to encode any of that object's information.
>
> > "Performance for baseline methods on the DAVIS dataset is rather low, comparing with state-of-the-art results."
>
> We use a subset of the Davis dataset in our experiments to ensure instances of each semantic class are present in both train and val sets: we consider 10 classes in common with COCO, resulting in 3401 training and 1558 validation frames versus the original 4209 and 1999 sets of frames. Thus the mIOU values we report are not directly comparable to the ones reported on the Davis 2019-val set, and that’s where the difference is coming from.
>
> > “... cannot find the demonstration of this statement.”
>
> In Table 2 we compare the effect of the location of the adversarial loss on the model's performance and show that placing it after the patch embedding gives better performance than at the end of the encoder. In Figure 2 we show how the adversarial loss is placed at the patch embedding level versus at the global level. We also show an example of the motion blur artifact in Figure 1.
>
> > “... typos …"
>
> Thank you for reading the paper closely. We have revised the text to make the word "labeled" consistent, break up run-on sentences, and fixed other typos.

---

### Official Review · Reviewer_79eQ · 2022-10-24

**Confidence:** 4
**Correctness:** 3
**Technical Novelty And Significance:** 3
**Empirical Novelty And Significance:** 3
**Recommendation:** 6

**Clarity, Quality, Novelty And Reproducibility:**

The method is well motivated and very clear. The motivation is perhaps a little unintuitive at first, but the method is simple (not a bad thing), and the results defend the story well.

**Strength And Weaknesses:**

Overall, the motivation of this work is quite interesting, and the idea that we can improve inference on videos due to the distributional shift is quite novel. The overall method is quite simple, at least in theory, although adversarial losses can be difficult to implement and train. The experiments show strong improvements over the baseline, and also demonstrate that this is not a general method for all segmentation networks (there must be a distributional shift), via the ablation.

Overall, this would be more compelling if more datasets were evaluated on. In addition to number, there are also 'video' datasets such as Cityscapes, where the images were recorded in a video manner, but do not seem to exhibit the same issues one might see in Davis. Would we expect to see similar improvements for those datasets? Or is there more clarification needed on the exact class of issues that we would expect this method to improve upon?

Another topic of interest is that this does not seem like a method that is limited to image -> video transfer, but should, in theory, work on any two sets of inputs from different domains (E.g. daytime -> night time). Would this method generalize to those cases?

**Summary Of The Paper:**

This paper presents a method to improve semantic segmentation performance on single images from videos, from a network trained on still images. The argument is that videos typically have several distribution shifts from standard images, such as framing and motion blur. Therefore, applying a pre-trained segmentation network trained on still images results in sub-par performance on videos. The proposed method is to apply an adversarial loss on the features from the segmentation network, where the discriminator tries to discern whether the input is from a still image or a video. This allows the network to learn the image distribution of videos. Experiments are performed with networks trained on COCO, and tested on the Davis dataset, using both CNN and transformer architectures. Ablations also show that the improvement is not evident when training and testing on still image datasets.

**Summary Of The Review:**

Overall, this is a nice simple method which shows strong improvements on a perhaps novel problem. I would be interested in knowing if this method would generalize to other distributional shifts (and perhaps more interested in why not, if so).

---

> ### Author Response · Authors · 2022-11-18
> **Clarification on image and video dataset characteristics and on generalization**
>
> We thank the reviewer for their comments.
>
> > "Overall, this would be more compelling if more datasets were evaluated on. In addition to number, there are also 'video' datasets such as Cityscapes, where the images were recorded in a video manner, but do not seem to exhibit the same issues one might see in Davis. Would we expect to see similar improvements for those datasets?"
>
> Thank you for the suggestion. The reason why we focused on Davis is because it shares more object classes with COCO than other video datasets. As long as we have a pair of image dataset and video dataset with common object classes, we believe the proposed approach is valid. Although other datasets like Cityscapes may display different types of motion (e.g., object motion vs. camera motion), the adversarial training would be able to minimize the distribution difference between objects in images and the target video dataset.
>
> For our method it is important that objects of the same class are consistently annotated between the image and video datasets; if this is not the case then it is possible for the discriminator to use the absence of a class to distinguish the domain, in which case the encoder learns not to encode any of that object's information. This is demonstrated in our findings testing our method on FBMS and COCO in Table 4, where the annotations for people and cars in FBMS are not very consistent.
>
> > "Another topic of interest is that this does not seem like a method that is limited to image -> video transfer, but should, in theory, work on any two sets of inputs from different domains (E.g. daytime -> night time). Would this method generalize to those cases?"
>
> In the paper we focused on the image-video domain gap because of the common practice of using image datasets in training video segmentation networks. We do agree that our losses would generalize to other domains because our approach does not rely on any features specific to images/videos. We have updated our discussion section with this point.

---

> > ### Comment · Reviewer_79eQ · 2022-12-08
> > **Thanks**
> >
> > Thank you to the authors for their responses to our reviews. In light of the valid concerns brought up by the other reviewers, I will modify my final review to a 6. I believe that their is merit to this work, but agree that there are issues that should be addressed before acceptance.

---

### Official Review · Reviewer_AueS · 2022-10-27

**Confidence:** 5
**Correctness:** 2
**Technical Novelty And Significance:** 2
**Empirical Novelty And Significance:** 2
**Recommendation:** 3

**Clarity, Quality, Novelty And Reproducibility:**

Clarity & Quality: The description is unclear at many points. 1. The problem definition is not well presented. It took me a long time to figure out whether the task is for semantic, or instance segmentation (i.e., it is figured out from (3)). This ambiguity leads to many questions when reading this draft: for instance segmentation, transferring from image to video domain needs to consider instance association, which is not mentioned anywhere; for semantic segmentation, using DAVIS is not ideal.  2. Many figures are blurry with too tiny texts, figure 1 is almost uninformative.

Novelty: This paper falls into the large category of adopting an adversarial loss to mitigating image-video domain gaps, that is investigated by tons of works in recent years. The idea is not new.

Reproducibility: the work seems reproducible according to the description.

**Strength And Weaknesses:**

Strength:

+ The paper is in general easy to understand, despite the ambiguity in task definition (see Clarity).
+ Ablations are performed and interesting founding is presented.

Weaknesses:

- Using adversarial losses to adopt image model to video domain is not new. The idea of clustered loss is also incremental.
- The claim on most self-supervised video representations focus on the recognition level, isn’t accurate. There are quite a lot of self-supervised video correspondence learning works that highly related to video segmentation task (i.e., most evaluate on the video segmentation task). [1] leverages self-supervised video correspondence technique to semi-supervised VIS. However, all above is not mentioned at all.
- Seems no STOA approaches are compared, and the baseline's performance is rather low.
- Davis is not a ideal dataset for semantic segmentation since most of the sequences contain very few objects, and almost one single object per category.

[1] Fu et al. Learning to Track Instances without Video Annotations. CVPR 2020.

**Summary Of The Paper:**

This paper performs video segmentation via labeled images and unlabeled videos, in which models in the image domain is adapted to the video domain via an adversarial loss. Ablations are conducted to determine at which level(s) the loss can be added to optimize the performance.

**Summary Of The Review:**

Based on the weaknesses and the various concerns discussed in the Clarity, Quality, Novelty And Reproducibility, I would not recommend acceptance per its current states.

---

> ### Author Response · Authors · 2022-11-18
> **Discussion on novelty and clarifying choice of datasets and baselines.**
>
> We thank the reviewer for their detailed comments.
>
> > "Using adversarial losses to adapt image model to video domain is not new."
>
> The reviewer brings up that there is a previous work using an adversarial loss to mitigate the image-video domain gap but to our knowledge we are the first to do so. Could the reviewer please provide a citation to the previous work that does this?
>
> > "[1] leverages self-supervised video correspondence technique to semi-supervised VIS. However, all above is not mentioned at all. [1] Fu et al. Learning to Track Instances without Video Annotations. CVPR 2021."
>
> Thank you for the reference to Fu et al., 2021. We have revised our related works section to include it. Our losses are orthogonal to the self-supervised correspondence learning explored in this video instance segmentation paper.
>
> > "Seems no SOTA approaches are compared, and the baseline's performance is rather low."
>
> Our goal in the experiments is to test whether our losses improve widely used networks in the difficult setting of no access to labeled videos. Both of our baseline networks are widely used models: for the 2D CNN we use DeeplabV3-Resnet101 and for the 2D/3D Transformer we use VideoSwin with 2D/3D window size. Without access to labeled videos the baselines perform poorly but we show that our losses can boost their performance.
>
> > "Davis is not an ideal dataset for semantic segmentation since most of the sequences contain very few objects, and almost one single object per category."
>
> Davis is a very widely used dataset with more shared classes in common with COCO than other video semantic segmentation datasets, such as Camvid or Cityscapes. We also provide results showing our method's benefits on FBMS, which often has multiple foreground objects labeled. Our work focuses on the object representations learned in video semantic segmentation. Objects are moving most in videos, rather than background classes, and we believe these class representations differ most depending on whether they are learned from image and video datasets.
>
> For clarity we have increased Figure 1's font size and Figure 2 overall.

---

### Official Review · Reviewer_SNu6 · 2022-10-28

**Confidence:** 3
**Correctness:** 4
**Technical Novelty And Significance:** 2
**Empirical Novelty And Significance:** 2
**Recommendation:** 3

**Clarity, Quality, Novelty And Reproducibility:**

As above, I think the novelty is limited. On the positive side it is quite clear.

**Strength And Weaknesses:**

Strengths:
- The idea of using clustering of features to avoid penalizing differing class frequencies in the two domains is a clever one. This is a nice idea for other domain adaptation problems as well.
- The end result is quite good performance on video segmentation.
Weaknesses:
- The novelty of this work is somewhat limited. It is mainly about applying existing domain adaptation approaches to this problem. This might be enough if the application itself was innovative or surprising, but I am not sure that is the case; the motivation seems very much to be that video frames and images are different domains.
- Is video segmentation only about image frame segmentation? I feel that motion-based cues should also play a role, and the fact that they don't seems to be more an issue with the dataset rather than the problem. In any case, I find the prospect of framing video segmentation as solely a different domain for image segmentation somewhat limiting.

**Summary Of The Paper:**

This paper proposes an approach for training a video segmenter by combining image segmentation datasets with unlabeled video datasets. The main innovation is to treat the problem effectively as a domain adaptation problem. As such, the paper uses an adversary to ensure that features from images and features from video frames are indistinguishable. An additional innovation is to perform this adversarial loss on combined clusters of features from videos and images so that the differing frequency of categories in image and video does not cause a problem

**Summary Of The Review:**

The paper proposes to use domain adaptation for video segmentation. While it has some clever ideas, I don't find this framing of video segmentation as domain adaptation particularly compelling or interesting enough to justify acceptance.

As a recommendation, I would suggest that the authors consider expanding the idea of clustered adversarial loss to more domain adaptation applications. I think that might make for a more valuable contribution.

---

> ### Author Response · Authors · 2022-11-18
> **Motivation and discussion on motion information**
>
> We thank the reviewer for their comments.
>
> > "The novelty of this work is somewhat limited. It is mainly about applying existing domain adaptation approaches to this problem. This might be enough if the application itself was innovative or surprising, but I am not sure that is the case; the motivation seems very much to be that video frames and images are different domains."
>
> Video segmentation is important for many applications, but there are not enough pixel-wise video annotations to train the models. Due to a lack of video segmentation annotations we are often forced to rely on image annotations. However, simply using image-pretrained representations for video tasks results in poor performance as shown by [1], and we need a way to best take advantage of the spatial representations from images for video models.
>
> What we present in this paper is a solution for this problem. Using an adversarial loss to mitigate the image-video domain gap has not been done in the past and our clustered loss is new.
>
> We explore this for the first time and are sharing our findings.
>
> > "Is video segmentation only about image frame segmentation? I feel that motion-based cues should also play a role, and the fact that they don't seems to be more an issue with the dataset rather than the problem. In any case, I find the prospect of framing video segmentation as solely a different domain for image segmentation somewhat limiting."
>
> We agree that motion information is vital for video segmentation, which is why we also tested applying our losses to a 3D Transformer model with windowed space-time attention. 3D Transformers are able to capture motion information using space-time self attention. Our results in Table 3 indicate that our losses can improve over the baseline 3D Transformer's performance.
>
> [1]: Kalogeiton et al. "Analyzing domain shift factors between videos and images for object detection." TPAMI, 2015.

---

### Decision · Program_Chairs · 2023-01-20

**Decision:**

Reject

**Justification For Why Not Higher Score:**

The concerns area critical and shared by the reviewers. No strong reason was found to overturn their recomendation.

**Justification For Why Not Lower Score:**

N/A

**Metareview: Summary, Strengths And Weaknesses:**

The reviewers shared common concerns on 1) the limited technical novelty and 2) insufficient experimental results.

On the novelty side, two reviewers commented that image-to-video domain adaptation using adversarial learning has been explored before. The authors rebutted that this criticism is incorrect, claimed their work is the first to do so, and asked the reviewers to provide references. Unfortunately, the reviewers did not provide the references but this meta reviewer was able to find a few relevant papers [A, B, C]. (This meta reviewer provided these references to the authors during the discussion phase, and later they admitted that this has been done but their application to segmentation task is unique. This is true, but it significantly undermines -- and invalidates -- their original claim which wasn't specific to segmentation.)

* [A] Andrew Kae and Yale Song. "Image to Video Domain Adaptation Using Web Supervision." WACV 2020. https://arxiv.org/abs/1908.01449
* [B] Jin Chen, Xinxiao Wu, Yao Hu, and Jiebo Luo. "Spatial-temporal causal inference for partial image-to-video adaptation." AAAI 2021
. https://ojs.aaai.org/index.php/AAAI/article/view/16187
* [C] Wei Lin, Anna Kukleva, Kunyang Sun, Horst Possegger, Hilde Kuehne, and Horst Bischof. "CycDA: Unsupervised Cycle Domain Adaptation from Image to Video." ECCV 2022. https://arxiv.org/abs/2203.16244

On the experiments side, the reviewers pointed out that the paper does not report SOTA performance on the dataset and compared to rather weak baseline approaches only. They also commented that evaluation on more compelling/larger benchmarks are necessary. Further, the reviewers acknowledged that the proposed approach seems general (not limited to image-to-video setting) and suggested evaluating on a more general domain adaptation setting.

Given these concerns, one reviewer downgraded their score from 8 to 6 during the discussion phase.

This meta reviewer carefully read all the reviews and the rebuttal, and found no strong reasons to overturn the reviewers' recommendation. We are recommending rejection in its current form of this submission.